# Completion of the Emergency Department “Big 6” in Patients with an Acute Hip Fracture Is Associated with a Lower Mortality Risk and Shorter Length of Hospital Stay

**DOI:** 10.3390/jcm12175559

**Published:** 2023-08-26

**Authors:** Nick D. Clement, Rose S. Penfold, Andrew Duffy, Krishna Murthy, Alasdair M. J. MacLullich, Andrew D. Duckworth

**Affiliations:** 1Edinburgh Orthopaedics, Royal Infirmary of Edinburgh, Little France, Edinburgh EH16 4SA, UK; andrew.duckworth@ed.ac.uk; 2Ageing and Health, Usher Institute, University of Edinburgh, Edinburgh EH25 9RG, UK; rose.penfold@ed.ac.uk (R.S.P.); a.maclullich@ed.ac.uk (A.M.J.M.); 3Lothian Analytical Services, NHS Lothian, Edinburgh EH4 2XU, UK; andrew.duffy@nhslothian.scot.nhs.uk; 4Department of Emergency Medicine, Royal Infirmary Edinburgh, Edinburgh EH16 4SA, UK; krishna.murthy@nhslothian.scot.nhs.uk; 5Department of Orthopaedics and Usher Institute, University of Edinburgh, Little France, Edinburgh EH16 4SA, UK

**Keywords:** Big 6, emergency department, fracture, hip, length of stay, mortality

## Abstract

The aims of this study were to assess whether completion of the emergency department (ED) Big 6 interventions (provision of pain relief, screening for delirium, early warning score (EWS) system, full blood investigation and electrocardiogram, intravenous fluids therapy, and pressure area care) in those presenting with an acute hip fracture were associated with mortality risk and length of acute hospital stay. A retrospective cohort study was undertaken. All patients aged ≥50 years that were admitted with a hip fracture via the ED at a single centre during a 42-month period were included. A total of 3613 patients (mean age 80.9; 71% female) were included. The mean follow up was 607 (range 240 to 1542) days. A total of 1180 (32.7%) patients had all six components completed. Pain relief (90.8%) was the most frequently completed component and pressure area assessment (57.6%) was the least. Completion of each of the individual Big 6 components, except for pressure areas assessment, were associated with a significantly (*p* ≤ 0.041) lower mortality risk at the 90-days, one-year and final follow-up. The completion of all components of the Big 6 was associated with a significantly (2.4 hours, *p* = 0.002) shorter time to theatre. Increasing number of Big 6 components completed were independently associated with a lower mortality risk: when all six were completed, the hazard ratio was 0.64 (95% CI 0.52 to 0.78, *p* < 0.001). Completion of an increasing number of Big 6 components was independently associated with shorter length of hospital stay and completion of all six was associated with a 2.3 (95% CI 0.9 to 3.8)-day shorter acute stay. The findings provide an evidence base to support the ongoing use of the Big 6 in the ED.

## 1. Introduction

It is well established that early surgery is associated with lower mortality and perioperative complication rates in patients with an acute hip fracture [1,2]. There is less evidence in relation to associations with preoperative interventions and outcomes [3,4], including interventions performed within the emergency department (ED) to optimise and prepare patients for surgery. The National Institute of Health and Care Excellence (NICE) guidelines state that a patient should have a pain assessment on presentation to the ED and hourly until admission to the ward [5]. The Scottish standards of care for hip fracture patients state that “patients who have a clinical suspicion or confirmation of a hip fracture should have the “Big 6” interventions/treatments before leaving the ED” [6]. The “Big 6” is composed of: provision of pain relief, screening for delirium, early warning score (EWS) system, full blood investigation and electrocardiogram, intravenous (IV) fluids therapy, and pressure area care [6,7]. Although these may be optimal for patient care, the associations of completion of these early interventions with patient outcomes have not previously been assessed. Due to the increased pressures on ED services which have been observed across the United Kingdom (UK) in recent times [8], completion of the “Big 6” in the ED has become more challenging due to increasing demands on medical staff [6]. Less than 50% of patients admitted to Scottish hospitals with an acute hip fracture had all components of the “Big 6” completed in the ED over the 5 years prior to 2021 [9].

The “Big 6” were designed to optimise patients for surgery earlier in the patient journey. This may result in a shorter time to surgery [10], which is associated with few complications, lower mortality, and shorter length of hospital stay [1,2]. However, it is not currently known whether time to surgery is shorter in patients with the “Big 6” completed. Furthermore, it is unknown if all parts of the “Big 6” demonstrate similar associations with patient outcomes. If some of the “Big 6” components have limited value, then the bundle components could be tailored to ensure optimal use of ED time and resources.

The primary aim of this study was to assess whether completion of the ED “Big 6” standard of care in patients with an acute hip fracture was associated with reduced mortality risk at 90 days following their admission. The null hypothesis was that there was no difference in 90-day mortality according to whether the components of the “Big 6” were completed. The secondary aims were to assess whether completion of the components of the “Big 6” were associated with: (1) mortality risk at 30-days, 1-year, and at final follow-up, (2) time to theatre, and (3) length of acute hospital stay.

## 2. Materials and Methods

### 2.1. Patient Population

This retrospective cohort study included all patients aged ≥50 years admitted with an acute hip fracture to a single, large orthopaedic trauma centre over a 42-month period (1 January 2019 to 30 June 2022). The study centre is the only trauma centre serving a population of approximately 850,000 and manages more than 1000 hip fractures annually. The inclusion criteria were patients with either an intracapsular or extracapsular (no more than five centimetres of distal extension from the lesser trochanter) fracture of the proximal femur, resident in the catchment area and presented via the ED. Patients with isolated fractures of the acetabulum, pubic ramus, greater trochanter, and periprosthetic fractures were excluded, as were patients not admitted through ED (therefore could not have the “Big 6” completed).

Patients were retrospectively identified from the local hip fracture database, with prospective data collected on a continuous basis as part of the national Scottish Hip Fracture Audit (SHFA). Patient demographics, fracture management, time of ED presentation and discharge, completion of the components of the Big 6, time to theatre, ASA grade, length of stay, and mortality were collected from electronic health records (EHR) (TrakCare, InterSystems Corporation, MA, USA) and contemporaneous documentation. ASA grade was obtained from the anaesthetic notes, recorded by a clinician at the time of surgery. Time in ED was calculated as the time of presentation to the time of discharge from ED. Completion of the “Big 6” components was prospectively collected by the local audit co-ordinator for the SHFA, and dichotomously recorded as a yes/no. Time to theatre was taken as per the SHFA guidelines, from time to admission to the ward to commencement of anaesthesia. These data were compiled by specialist local audit coordinators familiar with hip fractures and the trauma unit. The data were collated and assessed for completeness by a senior analyst (AD) as part of the routine activity of the SHFA. All data were handled in accordance with the UK Caldicott principles.

The Scottish Index of Multiple Deprivation (SIMD) was used to assign the socioeconomic status of each patient with assessment of seven domains: current income, employment, health, education, skills and training, housing, geographic access, and crime [11]. The current study used the updated SIMD rankings published in 2020 to assign a patient to a quintile of local data zone deprivations (1 = most deprived to 5 = least deprived) according to the patient’s postcode at the time of injury.

### 2.2. Outcomes

Acute length of stay (LoS) was defined as the number of days between admission to the ward to discharge from the trauma centre. The discharge destination was obtained from the regionwide EHR records. Patient mortality status was obtained from the local (study centre) hospital EHR which is the sole provider for national health care for the catchment population. This was conducted using each patient’s Community Health Index number (a national unique patient identifier).

### 2.3. Statistical Anlysis

The statistical analysis was performed using Statistical Package for Social Sciences (SPSS) software (IBM, Inc., Armonk, NY, USA) version 17 and was undertaken by the lead author (NDC). Independent Student’s *t*-tests and one-way analysis of variance (ANOVA) were used to assess the continuous variables (age, time in ED, time to theatre, length of stay) for significant differences between groups. A Pearson’s correlation test was used to assess the association of age, time in ED and time to theatre with length of stay. Categorical variables were assessed using a Chi square test for between-group comparisons (sex, SIMD, ASA grade, surgical management, and completion of the components of the “Big 6”). The Kaplan–Meier time to event methodology was used to assess patient survival. The log rank (Mantel–Cox) test was used to assess differences in survival between those that had components of the “Big 6” completed. Cox regression analysis was used to assess the independent association of factors with patient mortality when adjusting for confounding variables. Linear regression analysis was used to assess the independent association of factors influencing length of hospital stay when adjusting for confounding variables. A *p*-value of <0.05 was defined as statistically significant.

A power calculation was performed, based on the suggested reduction in mortality at 90 days defined for the Hip Attack study [12], which used a hazard ratio of 0.7 with an assumed background mortality risk of 13%. To achieve 80% power and using an alpha of 0.05 (two-tailed), a minimum of 3486 patients would be required for a 1:6 ratio (assumed completion rate): 581 versus 2905. Therefore, to achieve the number of patients required and the known admission rate for the unit, a 42-month study period was chosen.

## 3. Results

There were 3740 patients who presented to the ED with an acute hip fracture during the study period, of which 93 patients (2.5%) were from outside of the catchment population of the study centre and 34 (1%) were direct admissions to orthopaedics and were also excluded. The included study cohort consisted of 3613 patients sustaining a hip fracture of which there were 2564 (71.0%) females with an overall mean age of 80.9 (SD 10.1). The mean time in the ED, from presentation to admission to the ward, was 4.7 (SD 3.0) hours. The mean time to theatre, from admission to the ward, was 23.3 (SD 21.0) hours. The mean length of acute hospital stay was 11.6 (SD 10.0) days. The mean follow-up was 607 (SD 428, range 240 to 1542) days. During the follow up period, there were 1600 deaths identified.

### 3.1. Rate of Completion of the Big 6

There were 1180 (32.7%) patients that had all six of the “Big 6” components completed in the ED. The most frequently completed component was pain relief (90.8%) whereas pressure areas assessment (57.6%) was the least frequently completed (Table 1). Male patients and patients with a higher comorbidity (ASA grade 3 relative grade 2) were associated with not completing the “Big 6” in the ED (Table 2). Completion of all components of the “Big 6” was associated with a significantly (*p* = 0.002) shorter time to theatre (Table 2).

### 3.2. Mortality

Completion of each of the “Big 6” components, except for pressure areas assessment, were individually associated with significantly lower mortality risks at 90 days (primary outcome), one year, and at final follow up (Table 3, Figure 1). Male sex (*p* < 0.001), older age (*p* < 0.001), shorter length of time in the ED (*p* < 0.001), non-completion of the pain relief/delirium screen/NEWS/bloods and ECG/IV fluids (*p* < 0.001), greater ASA grade (*p* < 0.001), greater time to theatre (*p* < 0.001) and non-THA surgical management (*p* < 0.001) were all associated with a greater mortality risk following hip fracture (Table 4). After adjusting for confounding variables, completion of an increasing number of “Big 6” components was independently associated with a lower mortality risk, which was significant when two or more components were completed (Table 5).

### 3.3. Length of Stay

Increasing age, increasing ASA grade, fracture management other than a THA or cannulated screws, increasing time in ED, increasing number of completions of the “Big 6” (more specifically pain relief and blood and ECG), and greater time to theatre were all associated with an increased length of hospital stay (Table 6). After adjusting for these confounding factors, completion of an increasing number of “Big 6” components was independently associated with shorter length of hospital stay (Table 7): when all six components were completed, the length of acute stay was 2.3 (95% confidence intervals 0.9 to 3.8) days shorter.

## 4. Discussion

This study has demonstrated that completion of the “Big 6” in the ED for patients with an acute hip fracture was independently associated with a lower mortality risk, with completion of an increasing number of “Big 6” components being associated with a lower mortality risk. Completion of the “Big 6” was also associated with a shorter length of hospital stay; when four or more of the components of the “Big 6” were completed, there was a 2-day shorter length of hospital stay. However, all the “Big 6” components were completed in less than a third of patients presenting to the ED during the study period. Completion of pressure area assessment and the prescription of IV fluids were the components that were least likely to be completed. Male sex and increased morbidity, according to ASA grade, were factors associated with not completing all the “Big 6” components.

Multiple variables are associated with postoperative mortality following a hip fracture [13], with time to theatre being a potentially reversible factor [1,2]. The current study affirmed that male sex, increasing ASA grade, fracture management, and increasing time to theatre were independently associated with increased mortality risk following a hip fracture, which were demonstrated in a previous study at a study centre [14]. For every one-hour delay to theatre, for those going to theatre, there was a 0.3% increase in mortality risk; therefore, for each day delay to theatre the mortality risk was increased by nearly 7%. This is lower than that found by Welford et al. [1] who demonstrated a reduced risk of 0.86 at 30 days for patients undergoing surgery within 24 h. However, they dichotomised time to theatre as before and after a 24 h “cut off”, whereas the current study assessed time as a continuous variable. According to data from the current study, patients waiting an extra 48 h had an increased mortality risk of 14%, which is similar to the risk observed by Moja et al. [2] in a meta-analysis of patients undergoing surgery beyond 48 h. Nonetheless, even when adjusting for confounding factors including time to theatre, an increasing number of completed “Big 6” components was independently associated with a lower mortality risk. When all six were completed, the mortality risk was 36% lower. The effect may seem greater than anticipated for the completion of clinical standards of care, that, to the authors’ knowledge, do not have a strong evidence base to support their use other than representing best clinical practice [6]. The novel aspect of the current study was to demonstrate an independent association of the “Big 6” on patient mortality. There may also be other factors related to the rate and ability of the ED to complete the “Big 6” that may have influenced the mortality risk to the patient, such as stress on the healthcare system [15,16]. When the hospital is at maximum capacity and resources are stretched, this may result in less time being available in the ED to complete the “Big 6” and subsequently the care of patients through their journey beyond the ED may not have been optimal due to system pressures.

The hip fracture patient’s journey through the healthcare system begins with their admission to the ED, where the initial care is essential in ensuring optimisation for surgery [10]. One of the aims of the “Big 6” is to optimise the patient for their surgery [10], by identifying medical problems early and preventing deterioration while keeping the patient comfortable, which was shown in the current study with a shorter time to theatre by 2 h for patients with all components of the “Big 6” completed. A 2 h shorter time to theatre was shown to be independently associated with a 0.6% reduction in mortality risk for patients in this study. The completion of each component of the “Big 6” was also shown to result in a lower mortality risk at 90 days and beyond, except for pressure area assessments. However, the effect of pressure area care on mortality may not be measurable at one timepoint as the prevention of pressure ulcers occurs at every stage of the patient’s journey. Pressure ulcers are associated with significant morbidity for the patients, a higher rate of hospital readmission, and increased 30-day mortality [17]. Furthermore, in view of the recent increasing length of stay within the ED, with more than half of hip fracture patients waiting beyond 4 h for a hospital bed, pressure area care in the ED is essential to prevent a morbid pressure ulcer from occurring.

The incidence of hip fractures has increased by 22% in the Netherlands over the last two decades, which equated to a national healthcare cost of 425 million Euros annually [18]. This cost could be far higher if the observed length of hospital stay had not reduced from 14 to 7 days during the same time period. The current study showed a 2.3 day shorter length of hospital stay when all components of the “Big 6” were completed, which equates to more than a GBP 1400 [19] cost saving or an increased bed occupancy rate of 20%. However, failing to achieve the “Big 6” in ED may also be a marker of a health care system that is under strain to provide care not only in the ED but also throughout rest of the hospital due to scarce resources with a delay to discharge (also reflecting the strains on social care). It is also important to acknowledge the vital need for the interdisciplinary pathway required in hip fracture patients to optimise their outcomes and improve their functional recovery [20].

Limitations of the study include a single centre retrospective design, with a relatively small number of patients included compared to registry-based analysis. However, the study did achieve the numbers set out in the predefined sample size calculation for mortality reduction at 90 days. Furthermore, registry-based analyses may potentially not have the granularity of data available to allow for adjusting for confounding adequately which has been performed in the current study. Other factors and standards of care were not assessed in the current study. Further work should include a more detailed focus on the impact of the “Big 6” and whether additional care interventions, such as a nerve block performed in the ED, should form part of the evidence-based approach to hip fracture care to optimise the outcomes for these patients, often frail and at high risk of adverse outcomes [21]. Another limitation of the current study is that proactive clinical interventions may have been undertaken once a “Big 6” component was undertaken in the ED. For example, a patient with a score of 4 or more in the 4AT, suggestive of delirium, may have undergone further investigations and management of the underlying cause and been referred for specific management pathways to optimise them for surgery.

## 5. Conclusions

The completion of individual components of the “Big 6”, except for pressure area assessment, was associated with a lower mortality risk at 90 days and at one year in patients with a hip fracture. Increasing the number of completed components of the “Big 6” was independently associated with a lower mortality risk and shorter length of hospital stay, which supports their ongoing use in the ED to optimise the patients journey after their injury. Further work should be performed to understand why these associations exist, including any direct causal effects of the Big 6; for example, the potential prompt treatment of pain, delirium, or other potential explanations such as fluctuations in stress on the healthcare system.

## Figures and Tables

**Figure 1 jcm-12-05559-f001:**
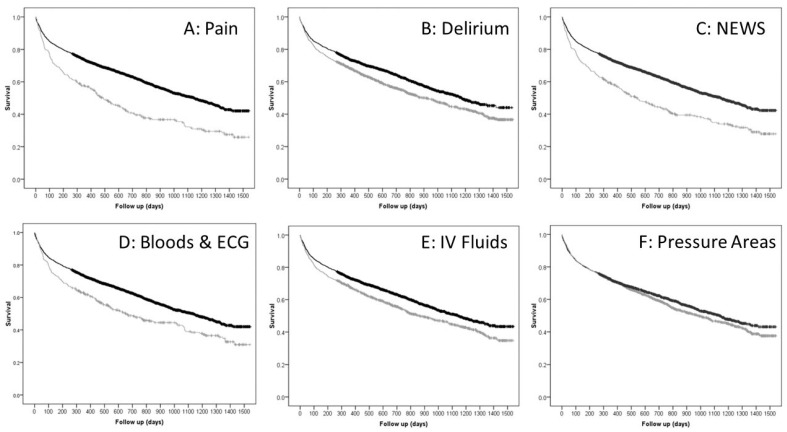
Kaplan–Meier curves for survival following a hip fracture according to completion (black line is for completed and grey line is not completed) for each component of the “Big 6”.

**Table 1 jcm-12-05559-t001:** The completion of the each of the Big 6 components for patient presenting to the Emergency department during the study period.

Component of the Big 6	Completed in ED (n, %)	
	Yes	No
Pain relief	3282 (90.8)	331 (9.2)
Delirium	2183 (60.4)	1430 (39.6)
NEWS	3235 (89.5)	378 (10.5)
Bloods/ECG	3201 (88.6)	412 (11.4)
IV Fluids	2560 (70.9)	1053 (29.1)
Pressure Areas	2080 (57.6)	1533 (42.4)

ED, Emergency Department; NEWS, National Early Warning Score; ECG, Electrocardiogram; IV, Intravenous Fluids.

**Table 2 jcm-12-05559-t002:** Patient demographics, ASA grade, fracture management, time in the ED, and time to theatre according to completion of the Big 6.

Demographic	Descriptive	Completion of the “Big 6”		Difference/ Odds Ratio (95% CI)	*p*-Value
		No (n = 2433)	Yes (n = 1180)		
Sex (n, % of group)	Male	737 (30.3)	312 (26.4)	Reference	
	Female	1696 (69.7)	868 (73.6)	1.21 (1.04 to 1.41)	0.017 *
Age (years: mean, SD)		80.8 (10.2)	81.2 (9.9)	0.4 (0.4 to 1.1)	0.328 **
SIMD (n, % of group)	1 (Most deprived)	276 (11.3)	129 (10.9)		0.345 *
	2	543 (22.3)	284 (24.1)		
	3	374 (15.4)	189 (16.0)		
	4	402 (16.5)	211 (17.9)		
	5 (Least)	835 (34.3)	366 (31)		
	Missing	3	1		
ASA Grade (n, % of group)	1	46 (1.9)	22 (1.9)	0.87 (0.52 to 1.47)	0.532 *
	2	476 (19.6)	313 (26.5)	1.20 (1.01 to1.43)	0.034 *
	3	1237 (50.8)	676 (57.3)	Reference	
	4	191 (7.9)	96 (8.1)	0.92 (0.71 to 1.20)	0.532 *
	Missing	483 (19.9)	73 (6.2)		
Fracture Management (n, % of group)	DHS	684 (28.1)	351 (29.7)		0.076 *
	Hemiarthroplasty	1088 (44.7)	545 (46.2)		
	IM Nail	293 (12.0)	141 (11.9)		
	Cannulated screw	141 (5.8)	55 (4.7)		
	THA	171 (7)	79 (6.7)		
	Other	56 (2.3)	12 (1)		
Time in ED (hours: mean, SD)		4.7 (3.0)	4.7 (3.0)	0.1 (−0.1 to 0.3)	0.568 **
Time to Theatre (hours: mean, SD)		29.2 (22.1)	26.8 (18.4)	2.4 (0.9 to 3.8)	0.002 **

* chi square test ** Student’s *t*-test. ED, Emergency Department; SIMD, Scottish Index of Multiple Deprivation; ASA, American Society of Anesthesiologists; SD, Standard Deviation.

**Table 3 jcm-12-05559-t003:** Patient survival following a hip fracture at different timepoints according to the completion of the component of the “Big 6” in ED.

Component of the “Big 6”	Completed in ED		Percentage Suvival (95% confidence interval)			
		N (%)	30 Days	90 Days	1 Year	Final
Pain Relief	Yes	3282 (90.8)	93.6 (+/−0.8)	85.9 (+/−1.2)	72.9 (+/−1.6)	42.9 (+/−2.7)
	No	331 (9.2)	91.5 (+/−3.1)	78.9 (+/−4.3)	57.0 (+/−5.3)	25.9 (+/−6.5)
	*p*-value *		0.148	<0.001	<0.001	<0.001
Delirium (4AT)	Yes	2183 (60.4)	93.9 (+/−1.0)	86.2 (+/−1.4)	73.6 (+/−1.8)	44.0 (+/−4.1)
	No	1430 (39.6)	92.7 (+/−1.4)	83.6 (+/−2.0)	68.1 (+/−2.4)	36.6 (+/−3.5)
	*p*-value *		0.215	0.033	<0.001	<0.001
NEWS	Yes	3235 (89.5)	93.8 (+/−0.8)	85.9 (+/−1.2)	73.0 (+/−1.6)	42.3 (+/−2.9)
	No	378 (10.5)	90.5 (+/−2.9)	79.6 (+/−4.1)	58.3 (+/−4.9)	27.9 (+/−5.9)
	*p*-value *		0.013	0.001	<0.001	<0.001
Bloods and ECG	Yes	3201 (88.6)	93.3 (+/−7.8)	85.7 (+/−1.2)	72.7 (+/−1.6)	41.9 (+/−2.9)
	No	412 (11.4)	93.9 (+/−2.4)	81.8 (+/−3.7)	61.7 (+/−4.7)	30.9 (+/−6.9)
	*p*-value *		0.650	0.041	<0.001	<0.001
IV Fluids	Yes	2560 (70.9)	93.7 (+/−1.0)	86.1 (+/−1.4)	73.1 (+/−1.8)	43.4 (+/−3.1)
	No	1053 (29.1)	92.5 (+/−1.6)	83.0 (+/−2.4)	67.5 (+/−2.7)	34.7 (+/−4.5)
	*p*-value *		0.184	0.014	0.001	<0.001
Pressure Areas	Yes	2080 (57.6)	93.1 (+/−1.2)	85.0 (+/−1.6)	71.4 (+/−2.0)	43.1 (+/−3.5)
	No	1533 (42.4)	93.9 (+/−1.2)	85.5 (+/−1.8)	71.6 (+/−2.4)	37.6 (+/−3.9)
	*p*-value *		0.363	0.623	0.906	0.105

* Log Rank (Mantel–Cox). ED, Emergency Department; NEWS, National Early Warning Score; ECG, Electrocardiogram; IV, Intravenous Fluids.

**Table 4 jcm-12-05559-t004:** Patient demographics, SIMD, ASA grade, time in ED, completion of each component of the Big 6 and completion of all components, fracture management, and time to theatre according to mortality following hip fracture of the study cohort.

Demographic	Descriptive	Alive (n = 2013)	Deceased (n = 1600)	Difference/ Odds Ratio (95% CI)	*p*-Value
Sex (n, % of group)	Male	511 (25.4)	538 (33.6)	Reference	
	Female	1502 (74.6)	1062 (66.4)	0.67 (0.58 to 0.78)	<0.001 *
Age (years: mean, SD)		78.3 (10.5)	84.2 (8.6)	6.0 (5.3 to 6.6)	<0.001 *
SIMD (n, % of group)	1 (Most)	234 (11.6)	171 (10.7)		0.097 *
	2	466 (23.1)	361 (22.6)		
	3	340 (16.9)	224 (14)		
	4	326 (16.2)	287 (17.9)		
	5 (Least)	645 (32)	556 (34.8)		
	Missing	2 (0.1)	1 (0.1)		
ASA Grade (n, % of group)	1	62 (3.1)	6 (0.4)		<0.001 *
	2	642 (31.9)	147 (9.2)		
	3	1014 (50.4)	899 (56.2)		
	4	84 (4.2)	203 (12.7)		
	Missing	211 (10.5)	345 (21.6)		
Fracture Management (n, % of group)	DHS	541 (26.9)	494 (30.9)		<0.001 *
	Hemiarthroplasty	868 (43.1)	762 (47.6)		
	IM Nail	227 (11.3)	207 (12.9)		
	Cannulated screw	134 (6.7)	62 (3.9)		
	THA	234 (11.6)	16 (1)		
	Other	9 (0.4)	59 (3.7)		
Time in ED (hours: mean, SD)		4.9 (3.2)	4.5 (2.7)	0.4 (0.2 to 0.6)	<0.001 **
Component of Big 6 Completed (n, % of group)	Pain relief	1891 (93.9)	1391 (86.9)	0.43 (0.34 to 0.54)	<0.001 *
	Delirium	1296 (64.4)	887 (55.4)	0.69 (0.60 to 0.79)	<0.001 *
	NEWS	1872 (93.0)	1363 (85.2)	0.43 (0.35 to 0.54)	<0.001 *
	Bloods/ECG	1831 (91.0)	1370 (85.6)	0.59 (0.48 to 0.73)	<0.001 *
	IV Fluids	1495 (74.3)	1065 (66.6)	0.69 (0.60 to 0.80)	<0.001 *
	Pressure Areas	1181 (58.7)	899 (56.2)	0.90 (0.79 to 1.03)	0.134 *
Completion of Big 6 (n, % of group)	Yes	683 (33.9)	497 (31.1)	Reference	
	No	1330 (66.1)	1103 (68.9)	0.88 (0.76 to 1.0)	0.068 *
Time to Theatre (hours: mean, SD)		27.2 (19.2)	30.1 (23.1)	3.9 (1.5 to 4.3)	<0.001 **

* Chi square test ** Student’s *t*-test. SIMD, Scottish Index of Multiple Deprivation; ASA, American Society of Anesthesiologists; ED, Emergency Department; NEWS, National Early Warning Score; ECG, Electrocardiogram; IV, Intravenous Fluids; DHS, Dynamic Hip Screw; IM, Intramedullary; THA, Total Hip Arthroplasty; SD, Standard Deviation.

**Table 5 jcm-12-05559-t005:** Cox regression analysis of variables associated with mortality following hip fracture for the study cohort.

Variable	Descriptive	Hazard Ratio	95.0% CI		*p*-Value
			Lower	Upper	
Sex (n, % of group)	Male	Reference			
	Female	0.67	0.60	0.74	<0.001
Age (years: mean, SD)		1.05	1.04	1.05	<0.001
SIMD	1	Reference			0.276
	2	0.89	0.72	1.09	0.259
	3	0.77	0.61	0.96	0.021
	4	0.94	0.76	1.17	0.580
	5	0.90	0.75	1.10	0.311
	Missing	0.00	0.00	N/A	0.861
ASA Grade (n, % of group)	1	0.32	0.14	0.71	0.005
	2	0.43	0.36	0.51	<0.001
	3	Reference			
	4	2.07	1.77	2.42	<0.001
	Missing	0.90	0.78	1.03	0.134
Fracture Management (n, % of group)	DHS	Reference			<0.001
	Hemiarthroplasty	0.98	0.87	1.12	0.812
	IMN	1.09	0.91	1.30	0.366
	Cannulated Screws	0.85	0.64	1.15	0.293
	THA	0.29	0.16	0.52	<0.001
	Other	2.48	1.39	4.44	0.002
Time to Theatre (hours: mean, SD)		1.003	1.001	1.005	0.017
Number of Big 6 completed in ED (n, % of group)	0	Reference			
	1	0.69	0.32	1.48	0.345
	2	0.53	0.32	0.88	0.014
	3	0.73	0.54	0.98	0.038
	4	0.65	0.52	0.82	<0.001
	5	0.60	0.49	0.74	<0.001
	6	0.64	0.52	0.78	<0.001

SIMD, Scottish Index of Multiple Deprivation; ASA, American Society of Anesthesiologists; ED, Emergency Department; DHS, Dynamic Hip Screw; IM, Intramedullary; THA, Total Hip Arthroplasty; CI, Confidence Intervals.

**Table 6 jcm-12-05559-t006:** Patient demographics, SIMD, ASA grade, time in ED, completion of each component of the Big 6 and number of completions, fracture management, and time to theatre and association with length of hospital stay following hip fracture of the study cohort.

Demographic	Descriptive	Length of Stay (Days)	*p*-Value
Sex (mean, SD)	Male	11.9 (10.0)	0.225 **
	Female	11.5 (10.0)	
Age (correlation coefficient)		r = 0.102	<0.001 *
SIMD (n, % of group)	1 (Most)	12.3 (11.6)	0.306 ***
	2	11.1 (9.1)	
	3	11.9 (9.7)	
	4	11.8 (10.3)	
	5 (Least)	11.5 (10.0)	
	Missing	4.7 (4.1)	
ASA Grade (n, % of group)	1	5.5 (3.8)	<0.001 ***
	2	9.7 (7.5)	
	3	13.1 (10.8)	
	4	13.1 (10.5)	
	Missing	6.1 (8.5)	
Fracture Management (n, % of group)	DHS	12.2 (10.3)	<0.001 ***
	Hemiarthroplasty	12.3 (10.4)	
	IM Nail	12.7 (9.4)	
	Cannulated screw	8.6 (8.0)	
	THA	6.4 (5.5)	
	Other	7.2 (6.8)	
Time in ED (correlation coefficient)		r = 0.101	<0.001 *
Pain relief	No	12.8 (10.1)	0.025 **
	Yes	11.5 (10.0)	
Delirium	No	11.8 (11.8)	0.231 **
	Yes	11.4 (9.7)	
NEWS	No	12.4 (10.4)	0.122 **
	Yes	11.5 (10.0)	
Bloods/ECG	No	12.9 (11.0)	0.007 **
	Yes	11.4 (9.9)	
IV Fluids	No	11.8 (10.8)	0.390 **
	Yes	11.5 (9.7)	
Pressure Areas	No	11.9 (9.7)	0.137 **
	Yes	11.4 (10.2)	
Completion of Big 6 (n, % of group)	0	13.8 (10.6)	0.005 ***
	1	12.3 (10.3)	
	2	8.8 (6.0)	
	3	11.1 (11.2)	
	4	11.3 (10.9)	
	5	11.6 (8.5)	
	6	11.4 (10.6)	
Time to Theatre (correlation coefficient)		0.153	<0.001 *

* Pearson’s Correlation ** Student’s *t*-test *** One-way Analysis of Variance. SIMD, Scottish Index of Multiple Deprivation; ASA, American Society of Anesthesiologists; ED, Emergency Department; NEWS, National Early Warning Score; ECG, Electrocardiogram; IV, Intravenous Fluids; DHS, Dynamic Hip Screw; IM, Intramedullary; THA, Total Hip Arthroplasty; SD, Standard Deviation.

**Table 7 jcm-12-05559-t007:** Regression analysis of variables associated with length of hospital stay following hip fracture for the study cohort.

Variable	Descriptive	Beta (Days)	95.0% CI		*p*-Value
			Lower	Upper	
Age		0.05	0.01	0.08	0.005
ASA Grade	1	Reference			
	2	1.05	−0.04	2.15	0.058
	3	3.23	2.27	4.19	<0.001
	4	2.60	1.15	4.05	<0.001
	Missing				
Fracture Management	DHS	Reference			
	Hemiarthroplasty	−0.02	−0.79	0.74	0.950
	IMN	0.43	−0.68	1.53	0.450
	Cannulated Screws	−3.11	−4.64	−1.59	<0.001
	THA	−4.11	−5.58	−2.65	<0.001
	Other	1.21	−3.94	6.35	0.646
Time in Emergency Department		0.17	0.06	0.28	0.003
Time to Theatre		0.04	0.03	0.06	<0.001
Number of Big 6 completed	0	Reference			
	1	0.48	−3.63	4.60	0.817
	2	−4.09	−6.85	−1.34	0.004
	3	−1.65	−3.52	0.23	0.085
	4	−1.85	−3.40	−0.30	0.019
	5	−1.88	−3.33	−0.43	0.011
	6	−2.34	−3.81	−0.88	0.002

ASA, American Society of Anesthesiologists; DHS, Dynamic Hip Screw; IM, Intramedullary; THA, Total Hip Arthroplasty.

## Data Availability

Data available upon reasonable request.

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
