# Peer review of "Completion of the Emergency Department “Big 6” in Patients with an Acute Hip Fracture Is Associated with a Lower Mortality Risk and Shorter Length of Hospital Stay"

_jcm, 2023, doi:10.3390/jcm12175559_

Round 1

Reviewer 1 Report

Review

Many thanks to the authors for having presented a so interesting original article about Completion of the Emergency Department “Big 6” in Patients 2 with an Acute Hip Fracture is Associated with a Lower Mortality Risk and Shorter Length of Hospital Stay”.

Before resubmitting the revision version of the article, please read the editorial rules carefully, and check other editorial aspects, such as: text alignment (lacking), text justification at the head (lacking), etc.

Title and Abstract

The title and abstract cover the main aspect of the work.

The abstract stands alone and it captures the appropriate essence of the manuscript.

The abstract is well structured, and it contains the main information of the study, but what “Big 6” are, it should be explained also in this part of the article.

Key words

Please provide them in alphabetic order: Big 6, fracture, emergency department, fracture, hip, mortality, outcome;

“Outcome” is too common to be a key word use trauma, or surgery.

Background

The introduction provides background and information relevant to the study.

The introduction identifies the problem that is being addressed in the manuscript, and clearly develops and states the purpose of the manuscript.

Methods

This section contains enough information to understand and possibly repeat the study.

There is an appropriate number of patients to justify the results.

Statistical analysis

The statistical analysis appropriate to the research.

Please provide who performed the analysis: an independent statistician or the same authors?

Results

The results presented are quite complete, reflecting the MM section.

The results are reproducible and reflective of clinical expectations, they are displayed in a readable fashion.

“Male patients and a higher ASA grade (grade 2 relative to 3) were associated with not 152 completing the “Big 6” in the ED”  pg 4 line 152  Don’t understand the values in the brackets in this sentence. Please specify better.

Discussion

The article presents an unbiased summary of the current understanding of the topic.

The manuscript presents a balanced view of recent work by active groups in the subject area.

The length and content of the discussion communicates the main information of the paper. However, the authors should discuss your results with other experiences in hip fracture patients, quoting:

·         Efficacy of an interdisciplinary pathway in a first level trauma center orthopaedic unit: A prospective study of a cohort of elderly patients with hip fractures. Arch Gerontol Geriatr. 2020 Jan-Feb;86:103957. doi: 10.1016/j.archger.2019.103957. Epub 2019 Oct 12. PMID: 31698279

This section recognizes the limitations of the manuscript.

The review contributes to the field.

You understood that Pressure Areas valuation in the ED doesn’t change the mortality significantly, why don’t you propose to remove that from the “big 6” since the ED is overloaded?

Conclusions

The conclusion provides a clear summary of the main point. It presents the significance of these points.

The conclusion is justified by the results and the methods.

References

The references are relevant to the study and in the correct style. They are up to date, but they should be integrated as suggested previously.

Competing interest

None of the authors' competing interests raise concerns about the validity of the article. 

Concerns

The manuscript or study does not raise any ethical concerns.

No concerns regarding similarities to other articles published by the same authors.

Tables and Figures

Tables are clear and legible; they are free from unnecessary modification.

The number and quality of tables are appropriate to transmit the main information of the paper.

Minor editing of English language required

Author Response

We would like to thank the reviewer for their constructive comments and we have tried to address these the best we can in a point by point response:

Review

Many thanks to the authors for having presented a so interesting original article about “Completion of the Emergency Department “Big 6” in Patients 2 with an Acute Hip Fracture is Associated with a Lower Mortality Risk and Shorter Length of Hospital Stay”.

Before resubmitting the revision version of the article, please read the editorial rules carefully, and check other editorial aspects, such as: text alignment (lacking), text justification at the head (lacking), etc.

Reply: I apologise for this, and we have tried to adhere to the editorial rules in the revised submission.

Title and Abstract

The title and abstract cover the main aspect of the work.

The abstract stands alone and it captures the appropriate essence of the manuscript.

The abstract is well structured, and it contains the main information of the study, but what “Big 6” are, it should be explained also in this part of the article.

            Reply: We agree with this suggestion and have now included this:

The aims were to assess whether completion of the emergency department (ED) Big 6 interventions (provision of pain relief, screening for delirium, early warning score (EWS) system, full blood investigation and electrocardiogram, intravenous fluids therapy, and pressure area care) in those presenting with an acute hip fracture were associated with mortality risk and length of acute hospital stay.

Key words

Please provide them in alphabetic order: Big 6, fracture, emergency department, fracture, hip, mortality, outcome;

“Outcome” is too common to be a key word use trauma, or surgery.

            Reply: We have changed as suggested and replaced outcome with length of stay.

Background

The introduction provides background and information relevant to the study.

The introduction identifies the problem that is being addressed in the manuscript, and clearly develops and states the purpose of the manuscript.

            Reply: No change. 

Methods

This section contains enough information to understand and possibly repeat the study.

There is an appropriate number of patients to justify the results.

Reply: No change. 

Statistical analysis

The statistical analysis appropriate to the research.

Please provide who performed the analysis: an independent statistician or the same authors?

            Reply: The analysis was performed by NDC. This has now been stated:

Statistical analysis was performed using Statistical Package for Social Sciences (SPSS) software (IBM, Inc., Armonk, New York, United States) version 17 and was undertaken by the lead author (NDC)

Results

The results presented are quite complete, reflecting the MM section.

The results are reproducible and reflective of clinical expectations, they are displayed in a readable fashion.

“Male patients and a higher ASA grade (grade 2 relative to 3) were associated with not 152 completing the “Big 6” in the ED”  pg 4 line 152  Don’t understand the values in the brackets in this sentence. Please specify better.

Reply: We are sorry for the lack of clarity here. The only difference in ASA was between grades 2 and 3. We have rewritten this and now reads:

Male patients and patients with a higher comorbidity (ASA grade 3 relative grade 2) were associated with not completing the “Big 6” in the ED (Table 2).

Discussion

The article presents an unbiased summary of the current understanding of the topic.

The manuscript presents a balanced view of recent work by active groups in the subject area.

The length and content of the discussion communicates the main information of the paper.

Reply: No change. 

However, the authors should discuss your results with other experiences in hip fracture patients, quoting:

Efficacy of an interdisciplinary pathway in a first level trauma center orthopaedic unit: A prospective study of a cohort of elderly patients with hip fractures. Arch Gerontol Geriatr. 2020 Jan-Feb;86:103957. doi: 10.1016/j.archger.2019.103957. Epub 2019 Oct 12. PMID: 31698279

Reply: We thank the reviewer for highlighting this reference and we have now included it in the discussion and now reads:

It is also important to acknowledge the vital need for the interdisciplinary pathway required in hip fracture patients to optimise their outcomes and improve functional their recovery.20      

This section recognizes the limitations of the manuscript.

The review contributes to the field.

You understood that Pressure Areas valuation in the ED doesn’t change the mortality significantly, why don’t you propose to remove that from the “big 6” since the ED is overloaded?

Reply: This is a controversial topic among the authors. The main reason this was included was to (1) document the pressures areas on admission of the patient and (2) to ensure these were cared for while awaiting admission to the word. Although we did not show and isolated effect of pressure area care when combined in the Big 6 – this was associated with a lower mortality risk and length of stay. We have included the justification in the discussion:

Completion of each component of the “Big 6” was shown to result in a lower mortality risk from 90-days and beyond, except for pressure area assessment. However, the effect of the pressure area care on mortality may not be measurable at one timepoint as prevention of pressure ulcers occurs at every stage of the patient’s journey. Pressure ulcers are associated with significant morbidity for the patients and a higher rates of hospital readmission and increased 30-day mortality.17 Furthermore, in view of recent increasing length of stay within the ED, with more than 75% of hip fracture patients waiting beyond 4-hours for a hospital bed, pressure area care in the ED is essential to prevent a morbid pressure ulcer from occurring. 

Conclusions

The conclusion provides a clear summary of the main point. It presents the significance of these points.

The conclusion is justified by the results and the methods.

Reply: No change. 

References

The references are relevant to the study and in the correct style. They are up to date, but they should be integrated as suggested previously.

Reply: No change. 

Competing interest

None of the authors' competing interests raise concerns about the validity of the article. 

Reply: No change. 

Concerns

The manuscript or study does not raise any ethical concerns.

No concerns regarding similarities to other articles published by the same authors.

 Reply: No change. 

Tables and Figures

Tables are clear and legible; they are free from unnecessary modification.

The number and quality of tables are appropriate to transmit the main information of the paper.

This is a very interesting manuscript with a good study design addressing the value of preoperative interventions on mortality in hip fracture patients. Nevertheless a few things should be addressed and corrected by the authors.

Reply: Again, we thank the reviewer for their comments and hope we have addressed these to satisfaction.

Reviewer 2 Report

This is a very interesting manuscript with a good study design addressing the value of preoperative interventions on mortality in hip fracture patients. Nevertheless a few things should be addressed and corrected by the authors.

1. Introduction line 68, sentence needs to be corrected.

2. The authors investigated patients with hip fractures. The high morbidity and mortality is typically seen in geriatric and frail patients. It is unclear why the authors included patients 50 yrs. And older instead of geriatric patients (<70yrs.). The reason for this should be explicitly addressed in the discussion.

3. It is unclear why the Big 6 were not completed in over 30% of the patient population. This presents a significant selection bias and should also be addressed in the discussion. The explanation given in the discussion that stress in the healthcare system resulted in lower completion rates of the big 6 in in sufficient and should be scrutinized by a separate analysis.

4. The authors state that fracture type did influence mortality. The type of fracture is not displayed in the manuscript, only the type of surgery. Given the higher mortality of intertrochanteric fractures the authors should separately analyze patients with intracapsular hip fractures and intertrochanteric fractures an compared these groups.

5. The different types of surgery were related to different rates in mortality with THA having the lowest mortality. THA is typically done in younger and active patients. It is therefor possible that the results were influenced by a selection bis. This should be addressed.

6. The was a significant difference in mortality in the DHS group. How do the authors explain this phenomenon?

7. In geriatric patients’ frailty is the most important determinant of mortality. Why was frailty not measured and included in the study?

8. The number of tables should be reduced.

Author Response

We would like to thank the reviewer for their constructive comments and we have tried to address these the best we can in a point by point response:

  1. Introduction line 68, sentence needs to be corrected.

Reply: We apologise for this error. This now reads:

Furthermore, it is unknown if all parts of the “Big 6” demonstrate similar associations with patient outcomes. If some of the “Big 6” components have limited value, then the bundle components could be tailored to ensure optimal use of ED time and resources.

  1. The authors investigated patients with hip fractures. The high morbidity and mortality is typically seen in geriatric and frail patients. It is unclear why the authors included patients 50 yrs. And older instead of geriatric patients (<70yrs.). The reason for this should be explicitly addressed in the discussion.

Reply: The inclusion of patients 50 years or more simply came from the Scottish Hip Fracture Audit inclusion criteria, which include patients 50 years and older. 95% were between 60 and 100 years old. We have not changed the analysis as inclusion of this patient should not have influenced the outcome measures assessed. Furthermore we did adjust for age in the adjusted analysis of outcomes.

  1. It is unclear why the Big 6 were not completed in over 30% of the patient population. This presents a significant selection bias and should also be addressed in the discussion. The explanation given in the discussion that stress in the healthcare system resulted in lower completion rates of the big 6 in in sufficient and should be scrutinized by a separate analysis.

Reply: We agree with the reviewer on this important point. We have therefore assessed factors associated with non-completion and identified male sex and increased (ASA 2 versus ASA 3) morbidity were associated with non-completion (Table 2). Furthermore, we have shown then failure to complete the big 6 was associated with significantly longer time to theatre (2.4 hours) and independently association with longer length of hospital stay and increased mortality risk.

  1. The authors state that fracture type did influence mortality. The type of fracture is not displayed in the manuscript, only the type of surgery. Given the higher mortality of intertrochanteric fractures the authors should separately analyze patients with intracapsular hip fractures and intertrochanteric fractures an compared these groups.

Reply: We did originally group all patients together – intra versus extracapsular fractures. However, we felt this lost the effect of younger fitter patients with minimally displaced intracapsular fractures and those “fitter” patients undergoing THA when compared to those undergoing a hemiarthroplasty. We therefore included the operation type in the models. We feel this is more granular data rather than grouping simply into intra and extracapsular factures.

  1. The different types of surgery were related to different rates in mortality with THA having the lowest mortality. THA is typically done in younger and active patients. It is therefore possible that the results were influenced by a selection bis. This should be addressed.

Reply: We agree that THA will be selected for the younger and fitter patient, which is therefore associated with a lower mortality risk. However, even when this this was included in the models – completion of the big 6 was associated with a lower mortalty risk and longer length of hospital stay.

  1. The was a significant difference in mortality in the DHS group. How do the authors explain this phenomenon?

Reply: In the unadjusted analysis this was the case but on adjusting for confound there was no difference in mortality with DHS vs Cannulated screws vs IM anil vs hemiarthroplasty (Table 5). Only THA was independently associated with a lower mortality rate, which as the reviewer suggest in comment 5 is like due to bias with only the more physical fit patients being offered a THA.

  1. In geriatric patients’ frailty is the most important determinant of mortality. Why was frailty not measured and included in the study?

Reply: Assessment of frailty can be done retrospectively and we did consider this, but felt it would not add to the clinical impact of the paper. We have included ASA grade (for those with it recorded) and feel this would be a marker of comorbidity and therefore overall frailty.

  1. The number of tables should be reduced.

Reply: We have retained the number of tables currently but would be happy to include table 6 as supplementary but we are not use how to upload this.

Round 2

Reviewer 1 Report

Manuscript has improved in this new version. This referee thinks that it could be suitable for publication. Congratulations to the authors.

Minor editing of English language required.

Reviewer 2 Report

They have sufficiently adressed my concernes. The paper is now suitable for publication.